# In Vitro Antimelanoma Properties of *Verbena officinalis* Fractions

**DOI:** 10.3390/molecules27196329

**Published:** 2022-09-26

**Authors:** Rabia Nisar, Sanjay Adhikary, Saeed Ahmad, Mohammad Abrar Alam

**Affiliations:** 1Department of Pharmaceutical Chemistry, Faculty of Pharmacy, The Islamia University of Bahawalpur, Bahawalpur 63100, Pakistan; 2Department of Chemistry and Physics, College of Sciences and Mathematics, Arkansas State University, Jonesboro, AR 72404, USA

**Keywords:** *Verbena officinalis*, melanoma, cytotoxicity, medicinal plant, LC-MS

## Abstract

*Verbena officinalis* is commonly used in traditional medicine to treat many ailments. Extracts of this plant are therapeutic agents for the potential treatment of different diseases, including colorectal and liver cancers, but have not been explored for their anti-melanoma potential so far. The goal of the current work was to prepare a methanolic extract and fractionate it using hexane, chloroform, ethyl acetate, butanol, and acetone to get semi-purified products. These semi-purified fractions were studied for their potency against melanoma cell lines. The three potent fractions (HA, VO79, and EA3) demonstrated 50% inhibition concentration (IC_50_) values as low as 2.85 µg/mL against the LOX IMVI cell line. All three fractions showed similar potency in inhibiting the growth of the B16 cells, a murine melanoma cell line. Based on high-resolution mass spectrometry (HRMS) data, for the first time, we report on lupulone A from this plant. LC-MS data also indicated the presence of hedergonic acid, serjanic acid, and other compounds in *V. officinalis* extracts.

## 1. Introduction

Humans have been utilizing plants therapeutically for a long time. The World Health Organization (WHO) estimates that almost 80% of the world’s population relies on medicinal plants to cover their basic healthcare needs [1]. Traditional medications made from plants are considered more effective and safer clinically when compared to synthetic entities [2]. Approximately 25% of the medications prescribed worldwide are originated from plants [3]. Medicinal plants containing a rich source of bioactive compounds are the most common source of novel drug discovery and are particularly useful as antimicrobial and anticancer therapeutics [4]. A recent analysis found that around 50% of approved anticancer drugs between 1940 to 2014 were obtained from natural sources or directly derived from them [5]. 

*Verbena officinalis*, a plant from the Verbenaceae family, is commonly called pigeons’ grass, herb of grace, or vervain. This plant is popularly called bitter herb or Kori-booti in Pakistan. It is mostly found in Asia, North Africa, and Europe. This plant is fairly distributed near water in cultivated fields and wastelands in the western and northern regions of Pakistan. It is a perennial erect herb, which grows to a height of about 25–100 cm, having toothed and lobed leaves. The delicate spikes hold elegant, silky, pale purple, or pink flowers [6]. In the traditional herbal system of medicines, *V. officinalis* has been employed for the treatment of many ailments such as gastric diseases, abrasion, skin burns [7], wounds, thyroid problems, rheumatic pain [8], asthma and cough [9], amenorrhea, enteritis, acute dysentery [10], expectorant and diuretic [11]. *V. officinalis* studied for its new important bioactivities including analgesic and anti-inflammatory [12,13], antioxidant [14], antifungal [15], anticonvulsant [6], antibacterial [16,17], anticancer [18,19], neuroprotective [10], antidepressant [20], antiproliferative [21], urolithiasis [22], and antitumor [23] effects. *V. officinalis* has a pool of bioactive metabolites, including flavonoids [24], sterols and triterpenoids [25], phenylethanoid glycosides [21], iridoids [26] and ursolic acid [27], which further explains the folk use of this plant [6]. In addition, the species nowadays is recognized as a valuable cosmetic plant, mainly due to the presence of essential oils. The vervain herb is characterized by high variability in chemical composition depending on its origin [28].

Melanoma is the most fatal and aggressive skin cancer that accounts for 3% of all malignant cancer [29]. Melanoma has a high potential for metastasis and invasion, which accounts for about 75% of death related to skin cancer worldwide [30]. It is the 5th and 7th most prevalent cancer in American men and women, respectively [31]. In the United States, the expected number of new cases and deaths in 2022 will be 99,780 and 7650, respectively [32]. Antimelanoma properties of plant extracts such as *Aloysia citrodora* essential oil inhibit melanoma cell growth and migration by targeting HB-EGF-EGFR signaling [33]. The aim of this study was to analyze the important secondary metabolites of *V. officinalis* for the evaluation of their antimelanoma potential. Until now, this is the first anti-melanoma study of *V. officinalis* extracts.

## 2. Results and Discussion

### 2.1. In Vitro Antimelanoma Studies of Different Fractions

Initially, we tested the different extracts of *V. officinalis* at 50 µg/mL against the LOX IMVI cells, a melanoma cell line. Acetone and n-butanol fractions did not show any cytotoxicity for this melanoma cell line at this concentration (Figure 1). Therefore, we did not pursue these two fractions of the methanolic extract of *V. officinalis* further. The extracts of hexane, chloroform, and ethyl acetate showed >70% growth inhibition at 50 µg/mL concentration. We further fractionated the extract of chloroform by column chromatography.

All these fractions were tested against LOX IMVI cell line at 50 µg/mL concentration. Extracts showing >70% growth inhibition were tested against the cell lines at five serial dilutions. We found three potent fractions (HA, VO79, and EA3) with 50% inhibition concentration (IC_50_) as low as 2.8 µg/mL against the LOX IMVI cell line (Table 1). The fraction VO79 inhibited the growth of melanoma cell lines (SK MEL 28, LOX IMVI, and SK MEL 5) with the IC_50_ values in the range of 6.2 to 11.6 µg/mL. This fraction inhibited the growth of a murine melanoma cell line (B16 cell line) with an IC_50_ value of 7.0 µg/mL. The HA fraction was found to be the most potent isolate of the *V. officinalis* to inhibit the growth of LOX IMVI cell lines. The EA3 fraction was very efficient in inhibiting the growth of SK-MEL-28 and LOX IMVI cell lines with the IC_50_ values of 4.8 and 3.3 µg/mL, respectively. All three fractions inhibited the growth of the B16 cell line effectively, with IC_50_ values as low as 6.2 µg/mL. These results are very significant as the positive controls is 3–4 times less potent than these fractions against the melanoma cell lines except for LOX IMVI cell line against cisplatin.

### 2.2. Phytochemical Analysis of V. officinalis by HR-ESI-MS

The three potent fractions were subjected to HR-EST-MS analysis. The data were compared with online databases, such as NIST Chemistry WebBook and PubChem. The structure of 13 metabolites of *V. officinalis* belonging to different phytochemical groups was putatively assigned using mass spectrometry (Table 2). We have found the bis(2-ethylhexyl)phthalate and lupulone A using high-resolution mass spectrometry (HRMS) data (Appendix A). This is the first report indicating the presence of lupolone A in this plant. This result is very significant, as lupulones are known to show anticancer properties [34]. Triterpenoids were the representative chemical class consisting of seven compounds—hederagonic or glycyrrhetinic acid (V4), (2*α*,3*β*)-2,3-dihydroxyurs-12-en-28-oic acid or hederagenin (V5), gypensapogenin A (V6), momordicinin (V8), camarolide (V9), ursonic or moronic acid (V10), and serjanic acid (V12). Three compounds (V3, V11, and V13) did not match the structures of the online database. 

## 3. Materials and Methods

### 3.1. Chemicals, Instruments, and Software 

Chemicals and silica: Methanol, n-hexane, chloroform, ethyl acetate, n-butanol, normal phase silica gel mesh size (70–230); Instrument: preparative TLC columns, Rotary evaporator, Shimadzu IT-TOF mass spectrometer, Cytation^TM5^ (BioTek, Winooski, VT, USA), and GraphPad Prism 9 (software).

### 3.2. Collection of Plant Materials

The whole plant of *V. officinalis* was collected during the flowering season, November 2017 to March 2018, from tehsil Gojra, district Toba Tek Singh Pakistan (Figure 2). The taxonomic status of the plant was verified by Dr. Zaheer-ud-Din Khan, Government College University, Lahore, Pakistan. A voucher number 3514 of the plant specimen was deposited in the department of Botany, Government College University, Lahore, Pakistan.

### 3.3. Extraction and Fractionation

A whole plant was shade dried and then ground into a coarse powder. Pulverized powder (10 kg) was macerated in 80% aqueous methanol for two weeks under normal conditions with occasional shaking. The methanolic extract was filtered and concentrated at 40 °C in a vacuum using a rotary evaporator under reduced pressure to yield a dry crude extract (460 g). The dry methanolic extract of *V. officinalis* was suspended in 1 L of distilled water and extracted with hexane, chloroform, ethyl acetate, n-butanol and acetone successively to obtain different solvent fractions. Each fraction was then concentrated by using a rotary evaporator and weighed. All extracts were then stored in sealed containers in a refrigerator for further purification and biological evaluation. 

#### 3.3.1. Hexane Fraction

After fractionating the methanolic crude extract using hexane solvent, 2 g of hexane fraction (HA) was produced as a yellow color oil.

#### 3.3.2. Isolation of EA3 from the Ethyl Acetate Fraction

The ethyl acetate fraction (100 g) was obtained by performing extraction of the crude methanolic extract with ethyl acetate solvent. A greenish powder was obtained from the ethyl acetate extract, which was recrystallized from methanol to obtain a white powder (EA3). 

#### 3.3.3. Isolation of VO79 from the Chloroform Fraction with Column Chromatography

The chloroform fraction (80 g) was subjected to column chromatography using a wet technique, and elution was started with hexane, chloroform, and chloroform-methanol solvent system. Elution was started at pure hexane, and the polarity was increased by 10% by adding chloroform until it reached 100% chloroform. The polarity was increased gradually by adding methanol with a 10% increment in the solvent system until 100% methanol. This whole process has resulted in the elution of 21 sub-fractions. The fractions eluted at the solvent system chloroform: methanol (9:1 to 7:3) was obtained in a significant quantity (11 g). It was again subjected to column chromatography. VO79 fraction was eluted with hexane: chloroform (1:9 to 0:1) solvent system with a 2% polarity increment. 

#### 3.3.4. N-Butanol Fraction

*V. officinalis* dried methanolic extract fractionated using n-butanol as the solvent, yielding a 95 g n-butanol fraction.

#### 3.3.5. Acetone Fraction

Dried methanolic extract was fractionated with acetone solvent, affording an acetone fraction (29 g).

### 3.4. Cytotoxicity Studies

The Resazurin assay was performed to assess the cytotoxicity of the compounds as we described previously [48,49]. A 96-well plate was seeded with 6000 cells/well and incubated for 24 h in the presence of growth media. After 24 h of incubation, cells were treated with six two-fold serial dilutions of fractions from 25 (µM) to 0.78 µM. Each dilution was treated in triplicate, and another 24 h incubation was performed. After 24 h of treatment, resazurin dye was added at a final concentration of 25 µg/mL, and again the cells were incubated for 4 h. Finally, the fluorescence intensity was measured at 560 nm excitation and 590 nm emission using Cytation^TM^ 5 (BioTek, Winooski, VT, USA). The non-linear regression of viable cells and treatment concentration was performed using a GraphPad Prism 9 (San Diego, CA, USA) to calculate the 50% inhibition concentration (IC_50_) of fractions.

## 4. Conclusions

For the first time, we have revealed the anti-melanoma properties of *V. officinalis* extracts. These mixtures of compounds are potent growth inhibitors in different melanoma cell lines with IC_50_ values at low micromolar concentrations. These potent fractions are very significant as these fractions of multiple compounds can be further separated into individual compounds, which can be significantly more potent than the mixtures. We have putatively characterized these fractions by using HRMS. The potent cytotoxic properties of these fractions warrant further separation and anti-melanoma studies of *V. officinalis* extracts.

## Figures and Tables

**Figure 1 molecules-27-06329-f001:**
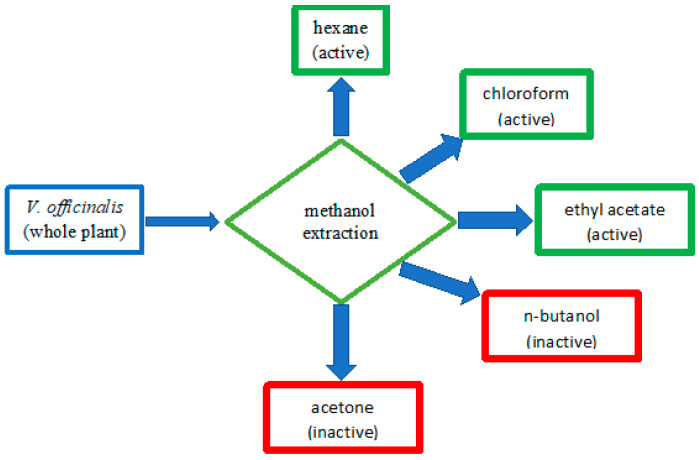
Flow chart for the melanoma active fractions of *V. officinalis*.

**Figure 2 molecules-27-06329-f002:**
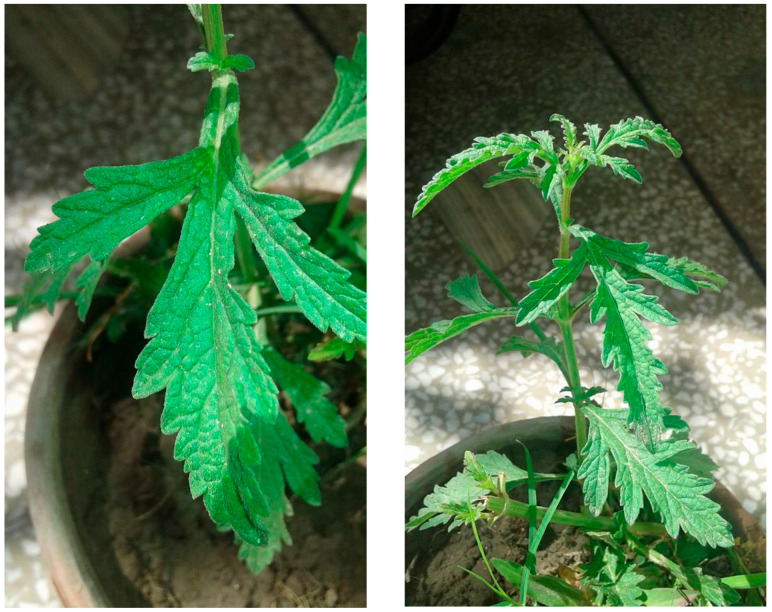
*Verbena officinalis* (06-21-2018, Chak NO. 363 JB, Tehsil Gojra, District Toba Tek Singh, Division Faisalabad 56000).

**Table 1 molecules-27-06329-t001:** IC_50_ values of the HA, VO79, EA3 fractions (µg/mL) against murine (B16) and human (SK MEL 28, LOX IMVI, and SK MEL 5) melanoma cell lines. NB: IC_50_ values are presented with mean ± Standard deviation. Taxol and cisplatin are positive controls.

Fraction	IC_50_ (µg/mL)			
	B16	SK MEL 28	LOX IMVI	SK MEL 5
HA	7.6± 1.1	10.6 ± 1.0	2.8 ± 0.2	8.0 ± 0.1
VO79	7.0 ± 0.8	7.2 ± 0.7	6.2 ± 0.1	9.6 ± 0.0
EA3	6.2 ± 0.8	4.8 ± 1.0	3.3 ± 0.0	6.2 ± 0.1
Taxol (µM)	27.4 ± 4.5	27.3 ± 2.81	32.1 ± 1.1	19.8 ± 3.1
Cisplatin	24.3 ± 1.0		5.4 ± 0.3	27.1 ± 3.1

**Table 2 molecules-27-06329-t002:** Tentative identification of the phytochemicals of *V. officinalis* extracts.

No		Probable Compounds	MF	*m*/*z* (found)	Class	Structure	*m*/*z* (cald)	Ref.
HA	V1	Bis(2-ethylhexyl) phthalate	C_24_H_38_O_4_	391.2812	Phthalate	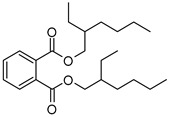	391.2842	[35]
	V2	Lupulone A	C_26_H_36_O_4_	413.2665	β-Bitter acids	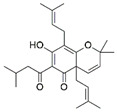	413.2686	[34]
Vo79	V3	Unknown	C_26_H_30_O_3_	391.23	-	-	-	-
	V4	Hederagonic acid or Glycyrrhetinic acid	C_30_H_46_O_4_	471.34	Triterpenoids		471.3468	[36,37]
	V5	(2*α*,3*β*)-2,3-Dihydroxyurs-12-en-28-oic acid or Hederagenin	C_30_H_48_O_4_	473.36	Triterpenoids		473.3625	[38] [39,40]
EA3	V6	Gypensapogenin A	C_30_H_42_O_2_	435.32	Triterpenes	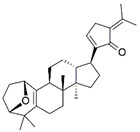	435.3257	[41]
	V7	Fistuloate A	C_30_H_44_O_2_	437.34	Aromatic compounds	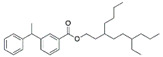	436.67	[42]
	V8	Momordicinin	C_30_H_46_O_2_	439.35	Triterpenoids	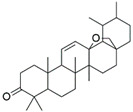	439.3570	[43]
	V9	Camarolide	C_30_H_44_O_3_	453.33	Triterpenoids	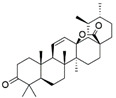	453.3363	[44]
	V10	Ursonic acid or Moronic acid	C_30_H_46_O_3_	455.35	Triterpenoids		455.3519	[45,46]
	V11	Unknown	C_31_H_48_O_4_	485.36	-	-	-	-
	V12	Serjanic acid	C_31_H_48_O_5_	501.36	Pentacyclic triterpenes	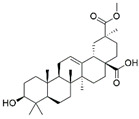	501.3574	[47]
	V13	Unknown	C_34_H_46_O_5_	511.34	-	-	-	-

## Data Availability

Not applicable.

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
