# Peer review of "In Vitro Antimelanoma Properties of Verbena officinalis Fractions"

_molecules, 2022, doi:10.3390/molecules27196329_

Round 1
Reviewer 1 Report
In the present study, authors obtained a methanolic extract, and corresponding fractionate of hexane, chloroform, ethyl acetate, butanol, and acetone. These semi-purified fractions were studied for their potency against melanoma cell lines. Three potent fractions (HA, VO79, and EA3) demonstrated lower inhibition concentration (IC50) values against four cell lines LOX IMVI, SK MEL 28, B16 and SK MEL 5 cell lines. In addition, LC-MS data was also provided. However, this paper cannot be published in Molecules. My individual comments were listed below.
1. Cell viability does not represent anticancer activity, more precisely, the cell viability was inhibited of the four cell lines. Therefore, more direct data on anticancer activity must to be added.
2. The chemical composition characterization data of mass spectrometry is not sufficient, and not only HA but also VO79, and EA3’s detailed chemical information should be supplied in Table 2.
3. The LC chromatogram and total ion chromatogram of the three fractions should be provided, other figures could be deleted.
Author Response
In the present study, authors obtained a methanolic extract, and corresponding fractionate of hexane, chloroform, ethyl acetate, butanol, and acetone. These semi-purified fractions were studied for their potency against melanoma cell lines. Three potent fractions (HA, VO79, and EA3) demonstrated lower inhibition concentration (IC50) values against four cell lines LOX IMVI, SK MEL 28, B16 and SK MEL 5 cell lines. In addition, LC-MS data was also provided. However, this paper cannot be published in Molecules. My individual comments were listed below.
- Cell viability does not represent anticancer activity, more precisely, the cell viability was inhibited of the four cell lines. Therefore, more direct data on anticancer activity must to be added.
We appreciate the reviewer for this comment. We think that preliminary cytotoxicity studies of these extracts are a good starting point for further antimelanoma studies, which will be reported in the future.
- The chemical composition characterization data of mass spectrometry is not sufficient, and not only HA but also VO79, and EA3’s detailed chemical information should be supplied in Table 2.
In this manuscript, we have reported the anti-melanoma properties of the extracts of V. officinalis. Based on the literature reports, we have tentatively assigned the structures of some of the compounds. We are not claiming the exact structure of the compounds.
- The LC chromatogram and total ion chromatogram of the three fractions should be provided, other figures could be deleted.
We have provided the LC chromatogram and the mass spectra. We think mass spectra are providing good information. Removing the mass spectra will justify the tentative structure of the compounds.
Reviewer 2 Report
The paper “Antimelanoma properties of Verbena officinalis fractions” presents research on the antitumor activity of V. officinalis extracts against melanoma. The article is interesting and the results are promising. However, there are several issues that require improvement or clarification before the work is accepted.
1. Publication concerning Aloysia citrodora antimelanoma activity could be added to the introduction.
2. A sentence from part 2.1. Chemicals & instruments can be divided into reagents, apparatus and software.
3. How much of the dry methanolic extract of V. officinalis was suspended in 1 L of distilled water before extraction with further solvents?
4. Please specify the type of neoplastic line in a sentence “Initially we tested the different extracts of V. officinalis at 50 μg/mL against the LOX IMVI cell line”.
5. In the sentence “Compounds showing >70% growth inhibition” “compounds” should be replaced with “extracts”.
6. Table 1 is not quoted in the manuscript. The values quoted in the text differ from the data from the table with the number of decimal places. Please standardize it.
7. In Table 2 it is worth mentioning in which fraction the given compounds occurred. What was the criterion of compounds order? Maybe better to sort them by molar mass or by the extracts that are present in?
8. The suggested structure of lupulone A differs from structure presented in ref [35]. There is no title in ref. [44] which makes it unreadable. Please correct also ref. 11 (first author's name).
9. The biggest concern is the identification of the potential components of the extract only based on the HRMS analysis. For example V4 could be glycerrythic acid, V5 hederagenin, V6 regelinol, V9 faradione, V11 ursoxy acid or moronic acid, while V13 of ursoxy acid. On the basis of what information were these compound structures selected?
10. Figure 13 or 14 (supplementary information) requires a correction of the retention time value.
Author Response
The paper “Antimelanoma properties of Verbena officinalis fractions” presents research on the antitumor activity of V. officinalis extracts against melanoma. The article is interesting and the results are promising. However, there are several issues that require improvement or clarification before the work is accepted.
- Publication concerning Aloysia citrodora antimelanoma activity could be added to the introduction.
We added this in the manuscript. Antimelanoma properties of plant extracts such as Aloysia citrodora essential oil inhibits melanoma cell growth and migration by targeting HB-EGF-EGFR signaling [33].
- A sentence from part 2.1. Chemicals & instruments can be divided into reagents, apparatus and software.
We divided 2.1 accordingly.
- How much of the dry methanolic extract of V. officinalis was suspended in 1 L of distilled water before extraction with further solvents?
480 g
- Please specify the type of neoplastic line in a sentence “Initially we tested the different extracts of V. officinalis at 50 μg/mL against the LOX IMVI cell line”.
We added “a melanoma cell line” at the end of the sentence.
- In the sentence “Compounds showing >70% growth inhibition” “compounds” should be replaced with “extracts”.
We changed it accordingly.
- Table 1 is not quoted in the manuscript. The values quoted in the text differ from the data from the table with the number of decimal places. Please standardize it.
We made the changes accordingly.
- In Table 2 it is worth mentioning in which fraction the given compounds occurred. What was the criterion of compounds order? Maybe better to sort them by molar mass or by the extracts that are present in?
We added an extra column to show the compounds from different extracts.
- The suggested structure of lupulone A differs from the structure presented in ref [35]. There is no title in ref. [44] which makes it unreadable. Please correct also ref. 11 (first author's name).
Rabia, address this
- The biggest concern is the identification of the potential components of the extract only based on the HRMS analysis. For example V4 could be glycerrythic acid, V5 hederagenin, V6 regelinol, V9 faradione, V11 ursoxy acid or moronic acid, while V13 of ursoxy acid. On the basis of what information were these compound structures selected?
We appreciate the reviewer for this comment to improve the quality of the manuscript. Wherever, different compounds are possible for the same molecular weight then we have considered any one of the possible structures. V4 could be hederagonic acid or glycerrythic acid, V5 hederagenin, V6 regelinol is different compound based on the molecular weight (M/z = 500)., V9 faradione, a different structure based on the molecular weight., V11 ursoxy acid or moronic acid, while V13 of ursoxy acid.
Round 2
Reviewer 1 Report
In the present study, authors obtained a methanolic extract, and corresponding fractionate of hexane, chloroform, ethyl acetate, butanol, and acetone. These semi-purified fractions were studied for their potency against melanoma cell lines. Three potent fractions (HA, VO79, and EA3) demonstrated lower inhibition concentration (IC50) values against four cell lines LOX IMVI, SK MEL 28, B16 and SK MEL 5 cell lines. In addition, LC-MS data was also provided. However, this paper cannot be published in Molecules in the present form. The author did not make additions and revisions exactly as suggested, my individual comments were listed below.
1. Cell viability on cell lines does not represent anticancer activity, related and direct data on anticancer activity must to be added.
2. The ion chromatogram of the three fractions has been provided, however, the LC chromatograms were still missed. Please provide. In addition, the chromatographic peaks in the LC should be marked and corresponding to the analyzed chemical structures in Table 2.
Author Response
- Cell viability on cell lines does not represent anticancer activity, related and direct data on anticancer activity must to be added.
We appreciate this comment from the reviewer. We are reporting in vitro preliminary studies of V. officinalis extracts and we believe that these preliminary data warrant publication as communication in Molecules. We have modified the title “In vitro Antimelanoma Properties of Verbena officinalis Fractions” and added “in Vitro” to make it more obvious for preliminary studies.
- The ion chromatogram of the three fractions has been provided, however, the LC chromatograms were still missed. Please provide. In addition, the chromatographic peaks in the LC should be marked and corresponding to the analyzed chemical structures in Table 2.
We get these data from outside sources and getting LC data will be time-consuming and it is costly.